# Emulsifying and Anti-Oxidative Properties of Proteins Extracted from Industrially Cold-Pressed Rapeseed Press-Cake

**DOI:** 10.3390/foods9050678

**Published:** 2020-05-25

**Authors:** Karolina Östbring, Kajsa Nilsson, Cecilia Ahlström, Anna Fridolfsson, Marilyn Rayner

**Affiliations:** Department of Food Technology, Engineering and Nutrition, Lund University, 221 00 Lund, Sweden; kajsa.nilsson@food.lth.se (K.N.); cecilia.ahlstrom@food.lth.se (C.A.); anna.fridolfsson@outlook.com (A.F.); marilyn.rayner@food.lth.se (M.R.)

**Keywords:** rapeseed press cake, protein recovery, emulsifying properties, anti-oxidative properties

## Abstract

One of the functional proteins in rapeseed—the amphiphilic protein oleosin—could be used to stabilize emulsions. The objectives of this study were to extract oleosins from cold-pressed rapeseed press-cake, optimize the extraction process, and investigate their emulsifying and anti-oxidative capacity. The proteins were recovered from industrially cold-pressed rapeseed press-cake at different alkali pHs. Emulsifying properties and oxidation rates were assessed. Oleosin extracted at pH 9 stabilized smaller emulsion droplets than oleosin extracted at pH 12, although the protein yield was higher at pH 12. Emulsions were formulated from flaxseed oil and corn oil and were stabilized by oleosin, bovine serum albumin, de-oiled lecithin and Tween 20 h and the emulsions were stored in accelerated conditions (30 °C) for 12 days. Oleosin stabilized emulsions to the same extent as commercial food-grade emulsifiers. Flaxseed oil emulsions stabilized by oleosin had a significantly lower concentration of malondialdehyde (MDA) which indicates a lower oxidation rate compared to BSA, de-oiled lecithin and Tween 20. For corn oil emulsions, oleosin and BSA had a similar capacity to delay oxidation and were significantly more efficient compared to de-oiled lecithin and Tween 20. Rapeseed oleosin recovered from cold-pressed rapeseed press-cake could be a suitable natural emulsifier with anti-oxidation properties.

## 1. Introduction

Formulated foods such as mayonnaise, salad dressings, and sauces are based on emulsions. Emulsions consist of two or more immiscible phases where one substance is dispersed in the other in the form of small droplets such as oil droplets-in-water (O/W) or water droplets-in-oil (W/O), where emulsion droplets need to be stabilized to prevent coalescence. Stabilization of food emulsions is often achieved through the addition of amphiphilic molecules such as small molecular weight surfactants (e.g., monoglycerides, polysorbates, and lecithin) or polymeric emulsifiers (e.g., dairy proteins, modified starches, and celluloses) [1], which act by decreasing the interfacial tension between the oil and water phases, and increase steric and/or electrostatic repulsion between emulsion droplets [2]. Proteins have been broadly utilized in all areas of food emulsion technology, where proteins from eggs and milk, are the most common [3]. The vast majority of plant protein-based emulsifiers are soy protein products (e.g., concentrates or isolates). However, because of allergen concerns the food industry is searching for other alternatives [4]. Thus, other types of plant-based proteins such as zein from corn [5], pea protein [6], and oleosin from oilseeds such as soy, canola, and rapeseed have been receiving increasing research interest [7,8,9,10].

Rapeseed (*Brassica napus*, *Brassica rapa* and *Brassica Juncea*, of rapeseed quality) is an important crop globally and is the second largest in oilseed production (after soybean), and the third-largest in vegetable oil consumption (after palm oil and soybean oil) [11]. In 2017, 76.2 million tons of rapeseed were harvested globally [12]. The crop is mainly used for the production of vegetable oils, which may be consumed directly or as an ingredient in food, or used as biofuel [13]. However, the oil extraction process leads to the generation of a solid residue amounting to 50–70% of the total quantity of seed processed (depending on the extraction method). This by-product is referred to as rapeseed press-cake (if cold-pressed) or rapeseed meal (if hot-pressed) and due to its high protein content (38–45%) it is used as a protein source in the livestock and aquaculture industries [14]. However, the direct application of rapeseed meal and press-cake is limited by its high content of fiber and anti-nutritional compounds, which can have a negative effect on the metabolism and growth performance of livestock [15]. In conjunction with the increased production of rapeseed worldwide, larger quantities of rapeseed meal and press-cake will be available [16]. At the same time, the interest in plant-based foods is growing, both from the consumers’ side and the food industry. There is a present need to recover and upcycle high value functional plant-based protein extracts with emulsifying and stabilizing properties. One such candidate is the structural protein oleosin from rapeseed. 

Rapeseed has a complex protein composition including both the storage proteins cruciferin and napin and oil body proteins such as oleosin and caleosin. Cruciferin and napin (11S globulin and 2S albumin) comprise 60% and 20%, respectively, of the total proteins in mature seeds, and oil body proteins accounts for the rest. Cruciferin has a molecular weight of 230–300 kDa and is composed of six sub-units each containing one acidic 30 kDa alpha chain and one basic 20 kDa beta chain which are linked by a disulfide bond [17]. Napin is a strongly basic protein with a molecular weight of 12–17 kDa [18] and is composed of a 4.5 kDa chain and a 9.5 kDa chain stabilized by disulfide bonds and disulfide bridges. The literature is reporting conflicting data on the isoelectric point (PI) of cruciferin and napin, but most commonly reported are PI 7.2 for cruciferin and PI of around 11 for napin [11]. The oil in rapeseeds is contained within oil bodies that are discrete intracellular structures consisting of an oil core, stabilized by a phospholipid monolayer embedded with proteins, of which 80–90% is oleosin [19] (Figure 1). The biological functions of oleosin are to prevent coalescence of the oil bodies within the seeds during periods of drought. Oleosins are 15–26 kDa in size, but generally not exceeding 18 kDa [8,20]. The protein structure of oleosin is the foundation of its function—both as a structural protein within the seed and as a potential emulsifier. Oleosin has three structural domains that assume a distinctive configuration at the oil-water interface: an amphipathic domain and the N-terminal (30 to 60 amino acid residues), a central hydrophobic domain (72 residues), and an amphipathic domain at the C-terminal (60 to 100 residues) [19]. The central hydrophobic domain is special with respect to the number of consecutive hydrophobic amino acid residues. In comparison to other emulsifying proteins (i.e., from dairy and eggs), oleosin has a somewhat large overall size, but a very large purely hydrophobic domain that allows strong adsorption at the oil-water interface (Figure 1) while the two amphipathic ends, together with phospholipids, create a compact interfacial layer. Taken together, the structural elements of oleosin can contribute to both physical and chemical stability in emulsions stabilized by them [21]. 

Several groups have reported on the use of oleosin proteins recovered from a variety of oilseeds including canola (rapeseed) meal and used them to stabilize oil droplets with and without phospholipids as co-surfactants [7,8,19,20,22,23,24,25,26,27]. In this approach oleosin protein, phospholipids and oil are mixed, creating emulsion droplets stabilized similarly to natural oil bodies [7,19]. Cruciferin has been reported to have high emulsifying capacity, although napin has been found to have a deteriorating effect on the emulsifying properties of rapeseed proteins [18]. This emulsifying capacity has made rapeseed proteins in general, and oleosin in particular, an interesting material to study as an emulsifier and thus rapeseed is an interesting plant-based alternative to other well studied protein emulsifiers. Commonly used protein emulsifiers are isolated from dairy and eggs, such as bovine serum albumin (583 residues, 66 kDa), ovalbumin (385 residues and 45 kDa) and β-lactoglobulin (162 residues and 18 kDa) [3]. 

In addition to emulsifying properties, oleosin from canola has also been found to substantially decrease the rate of lipid oxidation when used in combination with phospholipids [7]. This is important, as preventing lipid oxidation is a key challenge for food formulators and processors, since lipid oxidation has negative effects on food quality such as taste, appearance, texture, and shelf-life [28].

The objectives of this work were to extract oleosin from industrially cold-pressed rapeseed press-cake, optimize the extraction process, and investigate the emulsifying and anti-oxidative capacity of oleosin from industrially cold-pressed rapeseed. Most of the studies in the literature have been examining industrially hot-pressed rapeseed meal or defatted rapeseed meal on a laboratory scale. Since cold-pressed press-cake has not been exposed to high temperatures, the proteins are expected to be in their native state. This could be an advantage in terms of functional properties such as emulsifying capacity. To our knowledge there are no studies where the effect of alkali pH on functional properties such as emulsifying properties and anti-oxidative effect has been investigated with industrially cold-pressed rapeseed press cake as starting material. The overall aim is to increase the value of this by-product and to determine if the prepared oleosin precipitates can be a suitable replacement for conventional emulsifiers such as those obtained from dairy, egg, and soybean. 

## 2. Materials and Methods 

### 2.1. Sources of Materials and Chemicals 

Bovine serum albumin (BSA, A-3059, CAS 9048-46-8) were purchased from Sigma-Aldrich (St. Louis, MO, USA). De-oiled soy lecithin (Epikuron 100 PIP) were purchased from Cargill (Minneapolis, MN, USA). Tween 20 (polysorbate 20, 9005-64-5) USB (Cleveland, OH, USA). Citric Acid (C6H8O7, CAS 77-92-9), sodium chloride (NaCl, CAS 7647-14-5), sodium dihydrogen phosphate monohydrate (H2NaO4P·H2O, CAS 7558-80-7), disodium hydrogen phosphate dodecahydrate (Na2HPO4·12H2O, CAS 10039-32-4), trichloroacetic acid (TCA, CAS 76-03-9), and sodium hydroxide (NaOH, CAS 1310-73-2) was purchased from Merck (Darmstadt, Germany). 1,1,3,3-tetra methoxy propane (CAS 102-52-3), and 2-thiobarbituric acid (98%) (TBA, CAS504-17-6) were purchased from Sigma-Aldrich (St. Louis, MO, USA). Miglyol 812 was purchased from (Sasol AG, Brunsbüttel, Germany). Hydrochloric acid (HCl, CAS 7647-01-0) was purchased from VWR Chemicals (Fontenay sous -Bois, France). Fish oil (Möller’s Tran, Orkla Health, Solna, Sweden), flaxseed oil and corn oil (Zeta, Di Luca AB, Stockholm, Sweden) were purchased in the local supermarket. All other chemicals were of analytical grade. 

Cold-pressed rapeseed press-cake (*B. napus*) was a kind gift from Gunnarshögs Jordbruk AB, Hammenhög, Sweden, a producer of cold-pressed rapeseed oil. The oil temperature during extraction did not exceed 35 °C and no solvents were used during extraction. The press-cake was received at the beginning of the study and was stored in the freezer (−18 °C) until use. The moisture content in the press-cake was 10.3% and the composition on dry basis was as follows: protein content 30.4%, fat content 17.3%, carbohydrate content 12.1%, fiber content 33.5%, and ash content 6.7% (external analysis provided by Eurofins). 

### 2.2. Protein Extraction and Recovery from Cold-Pressed Rapeseed Press-Cake

The process scheme for protein extraction and recovery, essentially as described by Wijesundera [7], can be seen in Figure 2. Cold-pressed rapeseed press-cake was milled in a knife mill (Grindomix GM 200, Retsch, Germany) for 20 s and was thereafter soaked in tap water (1:10) to rehydrate for 2 h in ambient temperature. The mixture was homogenized (UltraTurrax T25, Staufen im Breisgau, Germany) at 13,500 rpm for 10 min followed by pH adjustment to 9–12 with 2 M NaOH. The homogenization step was repeated as described above and the dispersion was centrifuged (Beckman Coulter, Allegra^®^ X-15R Centrifuge, Brea, CA, USA) at 5000× *g*, 30 min, 25 °C. The supernatant from this separation step is hereafter referred to as the “liquid phase” and the precipitate as “solid phase”. Extraction coefficient was defined as:(1)$$\mathrm{Extraction}\;\mathrm{coefficient}=\frac{\mathrm{Mass}\;\mathrm{of}\;\mathrm{protein}\;\mathrm{in}\;\mathrm{liquid}\;\mathrm{phase}}{\mathrm{Mass}\;\mathrm{of}\;\mathrm{protein}\;\mathrm{in}\;\mathrm{rapeseed}\;\mathrm{press}-\mathrm{cake}}$$
and is a measurement of how large proportion of the protein could be solubilized during the extraction phase. The solid phase was discarded and the pH of the liquid phase was adjusted to 6.5 corresponding to the isoelectric point of oleosin in rapeseed [29]. Samples were stirred with a magnetic stirrer for 20 min and were thereafter centrifuged as described above. The supernatant from the second separation is hereafter referred to as “supernatant” and the precipitate as “precipitate”. Precipitation coefficient was defined as:(2)$$\mathrm{Precipitation}\;\mathrm{coefficient}=\frac{\mathrm{Mass}\;\mathrm{of}\;\mathrm{protein}\;\mathrm{in}\;\mathrm{the}\;\mathrm{precipitate}}{\mathrm{Mass}\;\mathrm{of}\;\mathrm{protein}\;\mathrm{in}\;\mathrm{the}\;\mathrm{liquid}\;\mathrm{phase}}$$
and is a measurement of how large a proportion of the protein available in the liquid phase that could be precipitated. The protein recovery yield was defined as:(3)$$\mathrm{Protein}\;\mathrm{recovery}\;\mathrm{yield}=\frac{\mathrm{Mass}\;\mathrm{of}\;\mathrm{protein}\;\mathrm{in}\;\mathrm{the}\;\mathrm{final}\;\mathrm{precipitate}}{\mathrm{Mass}\;\mathrm{of}\;\mathrm{protein}\;\mathrm{in}\;\mathrm{rapeseed}\;\mathrm{press}-\mathrm{cake}}$$
and is a measurement of how large a proportion of the protein in the starting material (rapeseed press-cake) that could be detected in the final rapeseed oleosin precipitate. Extractions were made in at least duplicates for each pH value.

Precipitates after the last centrifugation step were kept frozen in −18 °C for 24 h before freeze-drying in a laboratory freeze dryer (Hetosicc, Freeze dryer CD 12, Birkerod, Denmark). The plate temperature was 20 °C, the condenser −50 °C and the vacuum pressure of the dryer was 0.02 mbar. The residence time for the samples in the freeze dryer was 7 days. Freeze-dried rapeseed protein precipitates were used for all emulsion experiments and experiments evaluating the oxidation rate. 

### 2.3. Dry Matter Analysis and Protein Content

Dry matter content was determined according to the official method of analysis (AOAC 2007). The temperature was adjusted to 102 °C and analyses were performed in triplicate. 

Protein quantification was performed using a protein analyzer (Thermo Electron Corp., Flash EA, 1112 Series, Waltham, MA, USA). Each sample was ground using a mortar and pestle, approximately 25 mg were placed in a tin cylinder (diameter 30 mm) for analysis and aspartic acid was used as a reference. The conversion factor used was 6.25. Each sample was analyzed in triplicate. 

### 2.4. Emulsifying Properties of Emulsions Stabilized by Rapeseed Proteins

#### 2.4.1. Preparation of Emulsions 

Emulsions were prepared in at least duplicates in glass test tubes with 2 mL phosphate buffer (0.005 M, 0.2 M NaCl, pH 7.0), 1 mL medium chain triglyceride oil (miglyol oil) and varying concentrations of freeze-dried rapeseed protein precipitates extracted at different alkali pH. The emulsions were homogenized for 1 min at 22,000 rpm (Ystral, D-79282, Ballrechten-Dottingen, Germany). The emulsions were incubated at 4 °C for 1 h prior to further analysis. 

#### 2.4.2. Particle Size Analysis of Emulsions

The particle size distribution of emulsions droplets was analyzed by light scattering (Malvern, Mastersizer 2000 Ver 5.60, Worcestershire, UK). The dispersing unit was filled with MilliQ-water, and the pump speed was set to 2000 rpm. The glass tubes containing emulsions were turned upside down three times to allow representative sampling. The Refractive index (RI) of the sample was set to 1.45 (Miglyol) and 1.33 for the continuous phase (water). The absorption was set to 1.0 and the obscuration rate was 10–20%. Each emulsion was measured at least twice. 

#### 2.4.3. Emulsion Stability during Storage 

Emulsions were prepared in duplicates with 2 mL phosphate buffer (0.005 M, 0.2 M NaCl, pH 7.0), 1 mL fish oil and 8 mg (dry basis) of freeze-dried rapeseed protein precipitates extracted at pH 9 and 12, BSA and de-oiled soy lecithin, as described above (Section 2.4.1). To control microbial growth, sodium benzoate was added to the rapeseed precipitate at a concentration of 0.1% before freeze drying, since this is the maximum amount allowed in food applications according to the FDA [30]. Experiments performed in our laboratory showed no difference in emulsifying capacity after the addition of sodium benzoate, compared to before the addition (data not shown). Particle size distribution analysis was conducted every second day as described above and each emulsion was measured at least twice. The emulsions were stored in the dark at 30 °C for 12 days to investigate emulsion stability during storage in accelerated conditions using a food-grade oil. 

### 2.5. Oxidative Stability of Emulsions

Emulsions were prepared in glass tubes with 2 mL phosphate buffer (0.005 M, 0.2 M NaCl, pH 7.0), 8 mg freeze-dried rapeseed protein precipitate extracted at pH 9 or 12, and 1 mL flaxseed oil or corn oil. To produce a non-oxidized flaxseed oil, flaxseeds were cold-pressed in a screw press the same day as the production of emulsions and were kept dark and cold until onset of the experiments (i.e., for a few hours). BSA and Tween 20 were used as commercial controls and were prepared as above. BSA stabilized emulsions had 8 mg BSA (dry basis) and emulsions stabilized by Tween 20 had a concentration of 0.025 wt% in the continuous phase, to scale the emulsion droplet size to similar sizes as the emulsions stabilized by the other investigated emulsifiers. Since oxidation is a surface dependent reaction, the exposed area of the emulsions investigated should be of equal size (calculated as specific surface area, S = 6/*d*_32_). Two emulsions were prepared per oil and incubation day (16 emulsions in total per emulsifier and type of oil) and the emulsions were incubated at 30 °C for 16 days. Analysis of the lipid oxidation rate in the emulsions was performed every second day. 

To measure the rate of lipid oxidation, the Thiobarbituric acid reactive substances (TBARS) method was used; although the method is non-specific, it can give an indication of oxidative stability of different formulations [31]. The analysis aimed to quantify the amount of formed malondialdehyde (MDA) during storage in accelerated conditions. MDA reacts with thiobarbituric acid (TBA) and forms a pink compound that can be quantified at 532 nm in a spectrophotometer. 

Before the TBA-solution was added, the incubated emulsions were homogenized for 10 s at 22,000 rpm (Ystral, D-79282, Ballrechten-Dottingen, Germany) to allow representative sampling. 0.5 mL emulsion was quickly transferred to plastic tubes and 2.5 mL TBA-solution was added (0.375% (*w*/*v*) TBA, 15% (*w*/*v*) TCA and 0.25 M HCl). The blend was again homogenized for 10 s at 22,000 rpm (Ystral D-79282, Ballrechten-Dottingen, Germany) before the tubes were heated in a water bath (90–100 °C) for 10 min. The tubes were thereafter rapidly cooled under running water. Subsequently, the samples were centrifuged for 30 min at 25 °C 5000× *g* in a swinging bucket rotor (Beckman Coulter, Allegra^®^ X15R Centrifuge, Brea, CA, USA). The absorbance of the supernatant was quantified at 532 nm with a spectrophotometer (Varian Inc., Cary 50 UV-Vis, Palo Alto, CA, USA). The spectrophotometer was calibrated with TBA-solution. The analysis was performed in duplicate on each emulsion. A standard curve was constructed with 1,1,3,3-tetra methoxy propane in concentrations that ranged from 1–10 ppm, (y = 0.18x − 0.0176, R^2^ = 0.9971). The procedure for the analysis was the same as described above. From the standard curve, the equation to calculate the MDA concentration was obtained. The area under the curve was calculated using the trapezoidal method. 

### 2.6. Statistical Analysis

The area under the curve (AUC) was calculated for the emulsions and the concentration of MDA in oxidation trials by numerical integration using the trapezoidal rule method (Matlab R2020a). Data were analysed using SPSS software version 25 (IBM) and all data sets were normally distributed. Univariate general linear model with Tukey’s test was performed to investigate significant differences and results were considered significant if *p*-values were <0.05. All results were expressed as means with standard deviation.

## 3. Results and Discussion

### 3.1. Effect of Alkali pH on Protein Yield

Rapeseed meal has been reported to contain approximately 40–50% protein (dry basis) [32,33], whereas our starting material, the industrially cold-pressed rapeseed press-cake, had 28% protein (dry basis). The difference is explained by the fact that the press-cake used in this study had a higher oil content than that typically seen in conventional rapeseed and canola meal, which is hot-pressed, and extracted using hexane. It was observed that pH had a pronounced effect on total mass, solids, and protein mass (Table 1). At higher pHs during the extraction phase, solids were solubilized and detected in the liquid phase to a larger extent compared to extraction at lower pHs. The content of dry matter in the solid phase decreased with increased pH. The same phenomenon was observed for protein: with higher pH in the extracting medium, a larger amount of proteins could be extracted and detected in the liquid phase, thus resulting in a significantly higher extraction coefficient with increased alkalinity of the extraction media (Table 1 and Figure 3a). When pH of the extraction media was pH 9, 51% of the protein was solubilized and could be detected in the liquid phase after separation whereas, when pH was increased to 12, up to 72% of the proteins were solubilized in the liquid phase. This is a slightly higher extraction coefficient compared to a study conducted by Fetzer at al. where 59.5% of the rapeseed proteins could be extracted from cold-pressed rapeseed press-cake at pH 12. Ghodsvali et al. [34] reported that 35–60% of the proteins could be solubilized into the liquid phase after extraction and separation depending on alkali pH in the same range as in the present study [35]. Ghodsvali and colleagues used hot-pressed rapeseed press-cake, and we have previously reported that the absence of high temperatures during oil pressing was associated with higher protein recovery yields due to less degree of protein denaturation [36]. 

pH in the extraction media also had a significant impact on the precipitation step, where higher pH in the extraction phase resulted in a higher content of solids in the precipitate fraction. The same separation phenomenon was observed for protein with higher precipitation coefficient and higher protein recovery yield at extreme alkali pH during the previous extraction phase. When pH was adjusted to the isoelectric point of oleosin (6.5), only 38% of the available proteins could be precipitated and detected in the precipitate for samples extracted at pH 9, giving an overall protein recovery yield of 19% (Table 1 and Figure 3b). For samples where the extraction step was conducted at pH 12, 75% of the available proteins could be precipitated with a protein recovery yield of 54%. Rapeseeds contain both storage proteins and oil body proteins, where oil body proteins in the mature seed account for around 10% of the proteins. In all extractions in the present study, the recovery yield was well above 10%. Although oleosin has been reported to have an isoelectric point of 6.5 [7], storage proteins, i.e., cruciferin and napin, must have co-precipitated together with oleosin in the present study. Similar co-precipitation was reported by Wijesundera et al. [7] and both smaller protein structures with molecular weight of 7 kDa and 12 kDa (probably napin) and two protein structures with molecular weight of 30 kDa were reported. Cruciferin consists of subunits with alpha chains of 30 kDa and, due to the high yield, cruciferin is likely to be present in the precipitate. The most frequently reported isolecric point is 7.2 for cruciferin and around 11 for napin. Since the pH used for precipitation was 6.5 in the present study, most of the proteins in the precipitate are likely to be cruciferin and oleosin. With respect to the high isoelectric point, napin should be solubilized at pH 6.5 and, instead, be present in the supernatant fraction. 

Akbari et al., reported a recovery yield of 15–55% depending on alkali pH in the same range as in the present study although the process used was slightly different, with an acid washing step and several filtering steps included [17]. The pH of the extraction media in the present study affected not only the protein extraction from the starting material but also the subsequent precipitation process. pH 12 was therefore regarded as the best alkali extraction pH in terms of protein recovery yield. 

### 3.2. Emulsifying Properties

#### 3.2.1. Function of Protein Concentration and Extraction pH

All the rapeseed precipitates studied in this work were found to have the ability to stabilize emulsions. The mean droplet diameter of the produced emulsions decreased with increasing protein concentration for all extraction pH conditions, reaching a plateau of around 8 mg protein/mL oil (Figure 4a). Extraction pH affected the emulsifying properties of the rapeseed proteins where milder alkali treatment was associated with higher emulsifying ability. Proteins extracted at pH 9 and pH 10 stabilized significantly smaller emulsion droplets compared with proteins extracted at pH 11 and pH 12 in the investigated protein concentration interval (Figure 4b), although there were no differences in droplet size for emulsions stabilized by rapeseed precipitate extracted at pH 9 compared with pH 10, or pH 11 compared with pH 12. As the pH of the extraction medium increases from the protein’s isoelectric point, an increased charge repulsion drives the protein molecules from the native state to a partially unfolded state. The rapeseed precipitates in the present study includes a mixture of both oil body proteins and storage proteins where oil body proteins have been reported to have better emulsifying capacity than storage proteins, although cruciferin also contributes with some emulsifying ability. Globulins (cruciferin in rapeseed) have been reported to loose many side-chain interactions during exposure for extreme alkali conditions, altering the tertiary structure [37]. This disordered tertiary structure of the proteins could be the reason for the reduced emulsifying ability at extreme alkali pHs. 

The particle size distributions at lower protein concentrations (0.9–1.1 mg protein/mL oil) show one dominating peak around 65–92 µm representing emulsion droplets and one smaller peak around 2 µm representing aggregated proteins (Figure 4c). When the amount of protein in the formulation was increased to 14–19 mg protein/mL oil, smaller emulsion droplets (18–32 µm) could be stabilized (Figure 4d); however, larger emulsion droplets were still present to some extent. The smaller peak at around 2 µm had a higher magnitude which could be interpreted as a surplus of proteins in the continuous phase, not associated with the oil-water interface. The findings are generally in agreement with other studies found in the literature. However, since the starting materials, extraction methods and formulations differed it is difficult to make a direct quantitative comparison although emulsifying properties of rapeseed proteins as a phenomenon can still be discussed. The emulsifying properties of various types of oilseed isolates have been investigated by Chang et al. [4] who studied the emulsifying properties of protein isolates from pea, soy, lentil, and canola, and reported the droplet size of canola-stabilized emulsions to be around 14 µm (*d*_32_) when proteins were precipitated at pH 7. The precipitation pH in the study by Chang et al., was close to pH 6.5 used in the present study and the droplet sizes in our study varied between 7–12 µm (*d*_32_) depending on extraction pH, where lower pHs was associated with smaller emulsion droplet size. However, Chang et al., used a different extraction method with salt instead of alkali solution, and a filtering step was also included to remove the salt. Tan et al. [27] examined the emulsifying properties of proteins extracted from Australian canola meal using two different isolation techniques, an alkali extraction method with isoelectric precipitation similar to the present study, but also a salt-based extraction method (the Osborne method) were proteins are separated into globulins and albumins. Tan et al., reported that the albumin fraction stabilized emulsion droplets of around 17 µm (*d*_43_), the globulin fraction stabilized droplets of around 30 µm (*d*_43_) and the isoelectric precipitate stabilized significantly larger droplets, around 60 µm (*d*_43_). Tan et al. extracted the proteins at pH 12 and the pH during the precipitation was pH 4. However, the precipitated proteins in the present study stabilized emulsion droplets of 22–34 µm (*d*_43_), which is closer to the globulin fraction than the precipitate in the study by Tan et al. (Figure 4a). The most comparable extraction and precipitation method can be found in a study by Wijesundera et al., 2013, where pH was indeed 6.5, but the rapeseed used was defatted by solvents and not industrially cold-pressed as in the present study. The emulsions in the study had smaller droplet sizes (*d*_43_ 2–10 µm), but also somewhat higher concentrations of emulsifiers were used and, more importantly, the emulsions were homogenized at 1000 bar twice, which could explain the differences in droplet size between the present study and the study by Wijesundera et al. [7]. In the present study, a high shear homogenizer was used with a limit in the emulsion droplet size of 10 µm (*d*_32_). The emulsion model is therefore used for comparison between emulsifiers but does not investigate the lower limit of emulsion droplet size.

#### 3.2.2. Comparison with Other Emulsifiers during Storage

The particle size distribution of emulsions stabilized by freeze-dried rapeseed protein precipitates, BSA, and de-oiled lecithin was monitored under accelerated storage conditions (30 °C) for 12 days. There was a trend towards an increased mean droplet size which was significant for the de-oiled lecithin stabilized emulsion (Figure 5a). All emulsions exhibited creaming due to the relatively large size of the emulsion droplets (20–40 µm) produced at 8 mg/mL oil, but this did not result in coalescence or oiling off, with the exception of de-oiled lecithin which increased in droplet size by 40% over 12 days, and a layer of free oil was clearly visible (Figure 5b). Rapeseed proteins extracted at pH 9 stabilized smaller emulsion droplets during the 12 days stability test compared to proteins extracted at pH 12. These results were indicated already in the miglyol emulsion model (Figure 4a) but could be confirmed also in a fish oil emulsion system (Figure 5a), and the difference between the two rapeseed protein samples sustained also over a longer time.

### 3.3. Oxidative Stability 

The oxidative stability in emulsions over time was quantified as area under the curve (AUC) of the evolution of the oxidation product MDA (mg MDA/L emulsion). Selected types of food-grade emulsifiers from different groups (BSA (protein), de-oiled lecithin (phospholipid), and Tween 20 (non-ionic surfactant)) were used as the control for comparison with rapeseed protein precipitate (Figure 6). 

It was observed that for emulsions formulated using flaxseed oil as the dispersed phase, the oxidation rate was significantly reduced for rapeseed precipitates and BSA compared to other emulsifiers, especially in the case of Tween 20 (Figure 6a). In the emulsions formulated with corn oil, the overall oxidation rate was lower than in emulsions formulated with flaxseed oil due to less content of polyunsaturated fatty acids. Emulsions stabilized by rapeseed protein precipitates with corn oil as the dispersed phase had a reduced oxidation rate compared to emulsions stabilized by Tween 20 but could not be significantly differed from emulsions stabilized by BSA and de-oiled lecithin (Figure 6b). From the results, it was concluded that the emulsions stabilized by proteins, in general, had better stability against oxidation compared to lecithin and Tween 20. Tween 20 is a small surfactant and the concentration used in the emulsion formulation was scaled to match the droplet size of emulsions stabilized by rapeseed precipitates, BSA, and de-oiled lecithin. Proteins are large molecules compared to phospholipids and surfactants and they can physically cover a larger area of the oil-water interface, thereby hindering access to the substrate for oxidative agents. The rapeseed protein oleosin also has a structure tailored to stabilize the oil-water interface. Oleosin consists of two amphipathic domains associated to the oil surface on either side of the fully hydrophobic domain anchoring into the oil drops. In the two amphipathic domains, positively charged amino acid residues face the negatively charged lipid on the boundary with the phospholipid layer, while the negatively charged residues face outwards into the continuous phase [19]. This arrangement of oleosin protein at the oil-water interface is suggested to act as a barrier towards oxygen and reactive hydroperoxides, thereby protecting components within the oil droplets [38]. Wijesundera et al., reported similar results when the oxidative stability of rapeseed proteins (oleosins) were investigated in an emulsion system with tuna oil as the dispersed phase [7]. Oleosins had superior oxidative stability compared to the surfactant Tween 40, although another method of analysis was used so no direct comparison could be made with the present study. 

When extracting and precipitating proteins, other protein species are inevitably co-extracted. Wijesundera et al., investigated whether it was the oleosins or the co-extracted storage proteins in the rapeseed that was responsible for oxidative stability. They quantified the oxidative stability of emulsions stabilized by oleosin extract and commercial canola protein isolate (where napin and cruciferin are the dominating species), and the oleosin-rich material had superior oxidative stability. Although no protein characterization was conducted in the present study, it is likely that the oleosins with their unique protein structure are mostly responsible for oxidative stability superior to de-oiled lecithin (in flaxseed oil) and Tween 20 (in both flaxseed oil and corn oil).

## 4. Conclusions

The pH under alkali extraction had a profound effect on the protein recovery yield, and higher pH during the leaching phase was associated with higher yields after precipitation. However, extreme alkali conditions had a reducing effect on the proteins’ emulsifying properties and rapeseed proteins extracted at pH 9 had better emulsifying properties compared to proteins extracted at pH 12. Rapeseed proteins extracted at pH 9 had a significantly higher oxidative stability compared to de-oiled lecithin and Tween 20 in emulsions formulated by flaxseed oil. Rapeseed proteins recovered from cold-pressed rapeseed press-cake could, therefore, be a suitable natural emulsifier with an anti-oxidative effect. 

## Figures and Tables

**Figure 1 foods-09-00678-f001:**
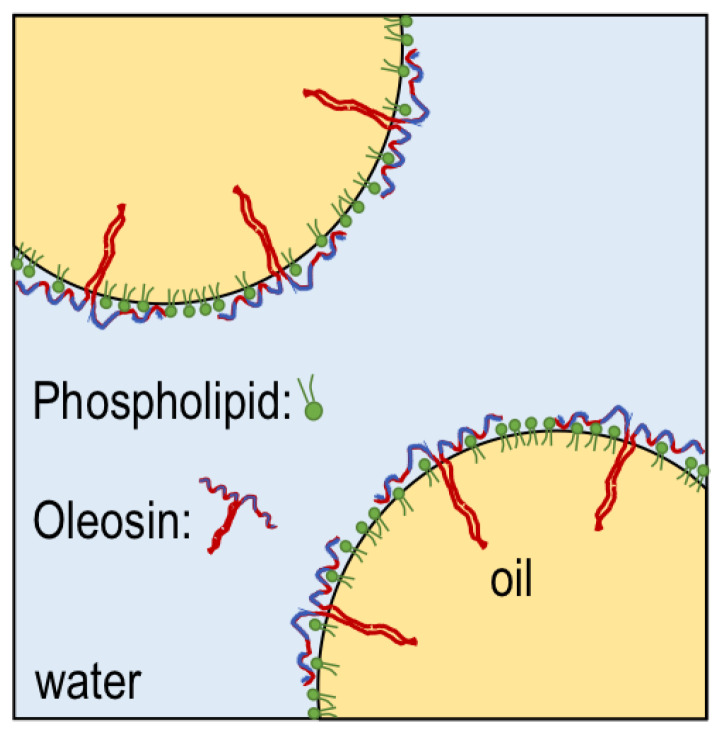
Illustration of oleosin proteins stabilizing oil bodies and their conformation at the interface. Oleosins are absorbed both at hydrophilic regions (purple) as well as at a central hydrophobic pin (red). Phospholipids are indicated in green. Illustration adapted from M. Rayner [21].

**Figure 2 foods-09-00678-f002:**
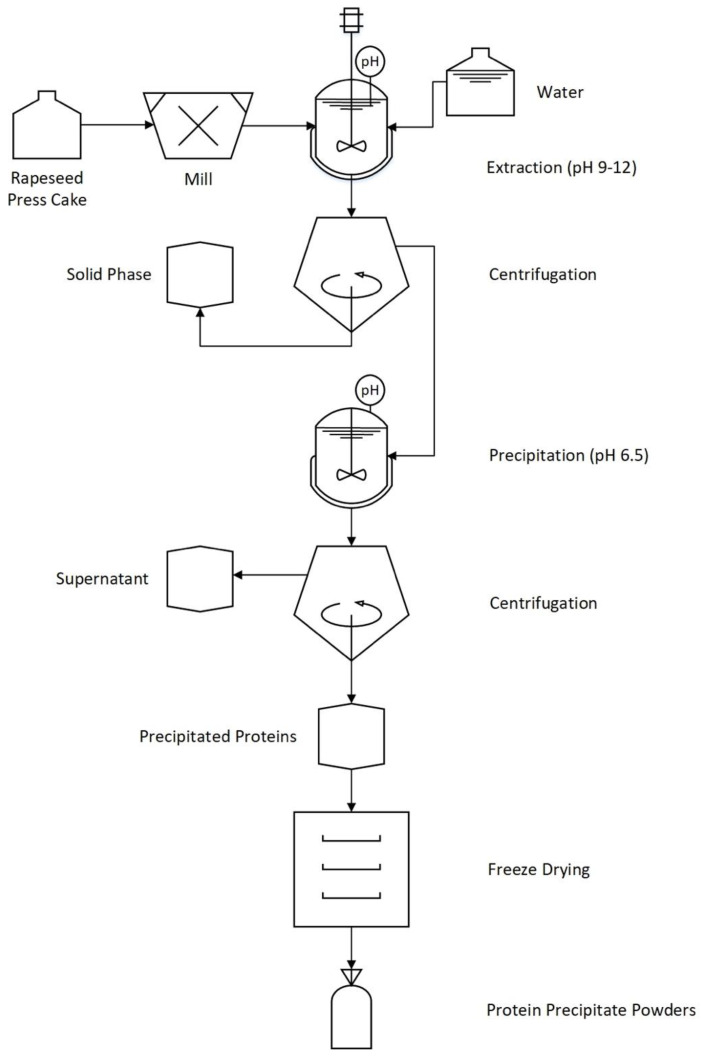
Schematic illustration of the rapeseed protein extraction process.

**Figure 3 foods-09-00678-f003:**
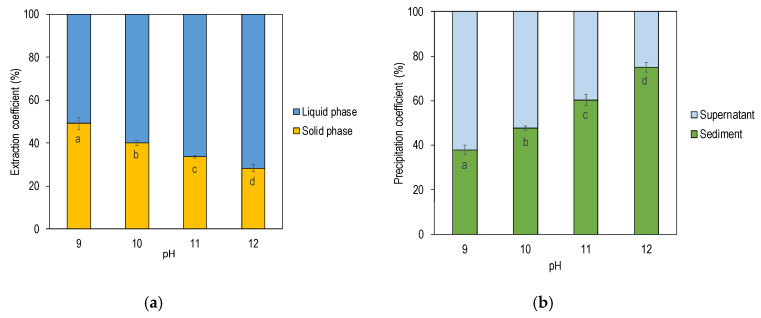
(**a**) Extraction coefficient displaying how rapeseed protein is distributed in the liquid phase and solid phase respectively after alkali extraction and separation. (**b**) Precipitation coefficient displaying how rapeseed protein is distributed after precipitation and separation. Data is an average of three extractions at each pH value. Different letters within each figure indicate significant difference *p* < 0.05.

**Figure 4 foods-09-00678-f004:**
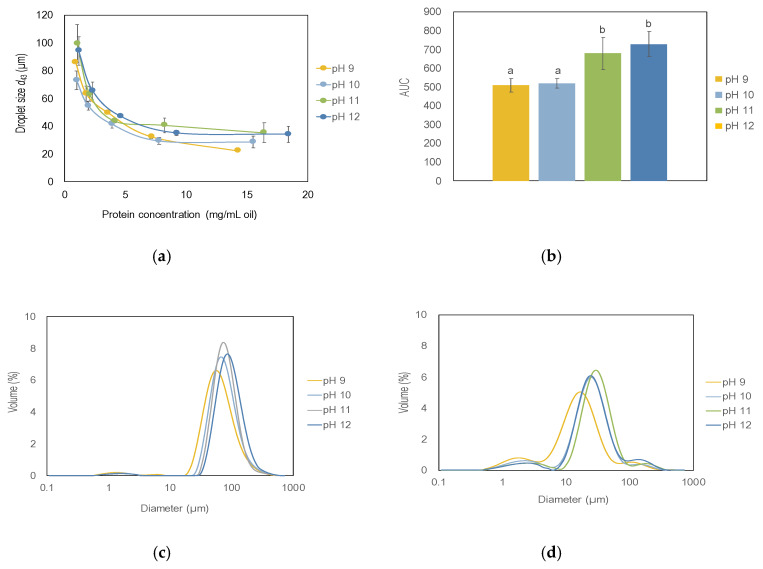
(**a**) Emulsion droplet size (d_43_) as a function of protein concentration in emulsions stabilized by rapeseed protein extracted at different pHs. Emulsions were 30% oil-in-water emulsions produced by high shear homogenization. (**b**) Area under the curve for droplet size *d*_43._ Different letters indicate significant differences, *p* < 0.05. (**c**) Size distributions of emulsions stabilized by rapeseed protein extracted at different alkali pH values. Protein concentrations in the emulsions were 0.9–1.1 mg/mL oil. (**d**) Size distributions of emulsions stabilized by rapeseed protein extracted at different alkali pH values. Protein concentrations in the emulsions were 14–18 mg/mL oil. Data is an average from six measurements of two extractions at each pH value.

**Figure 5 foods-09-00678-f005:**
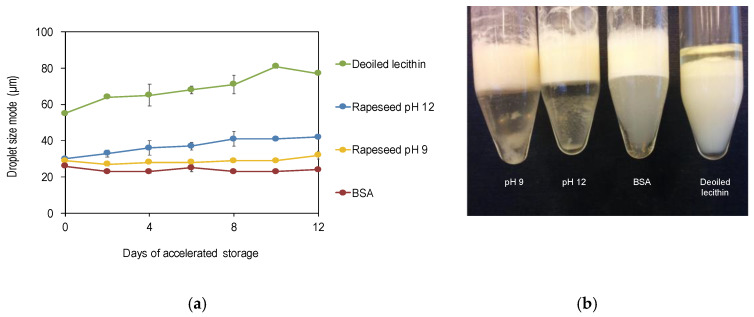
(**a**) Droplet size evolution of fish oil emulsions (mode) incubated at 30 °C for 12 days. Data is an average of six measurements from two replicates of emulsions. (**b**) Emulsions with fish oil after 12 days of storage at 30 °C. pH 9 is rapeseed protein extracted at pH 9, pH 12 is rapeseed protein extracted at pH 12, BSA is bovine serum albumin. 8 mg protein/mL oil (8 mg dry matter of de-oiled lecithin) was used to formulate 30% oil-in-water emulsions.

**Figure 6 foods-09-00678-f006:**
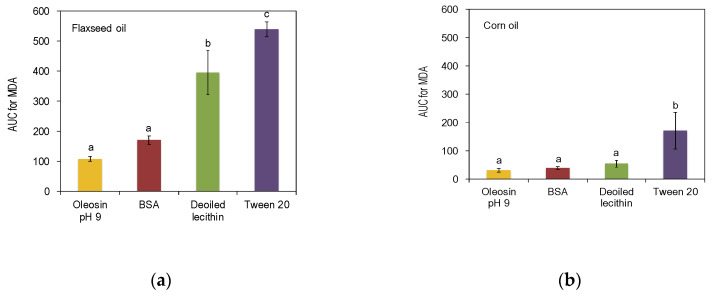
Oxidative stability for (**a**) flaxseed oil emulsions (**b**) corn oil emulsions stabilized by rapeseed protein extracted at pH 9, bovine serum albumin (BSA), de-oiled lecithin and Tween 20 expressed as AUC for malondialdehyde (MDA) concentrations (mg MDA/L emulsions). Data is an average of four measurements of two emulsion replicates for each oil. Different letters indicate significant differences within each figure, *p* < 0.05.

**Table 1 foods-09-00678-t001:** Mass balance over the protein recovery process with different alkali pH in the extraction phase. The starting material had a mass of 50 ± 0 g for each batch, the dry matter was 46 ± 0 g and protein content was 14 ± 0 g. Data are expressed as mean ± standard deviation. Different letters in each row indicate significant differences, *p* < 0.05.

		pH 9	pH 10	pH 11	pH 12
Centrifugation After Meal Extraction	**Liquid phase**				
	Mass (g)	360 ± 1 ^a^	370 ± 4 ^b^	370 ± 1 ^b^	370 ± 1 ^b^
	Dry Matter (g)	18 ± 0 ^a^	20 ± 1 ^b^	21 ± 0 ^b^	22 ± 1 ^c^
	Protein (g)	7.2 ± 0.3 ^a^	8.5 ± 0.2 ^b^	9.5 ± 0.1 ^c^	10 ± 0.2 ^d^
	Protein Yield (% of Starting Material)	51 ± 0 ^a^	60 ± 0 ^b^	66 ± 0 ^c^	72 ± 0 ^d^
	**Solid-Phase**				
	Mass (g)	134 ± 1 ^a^	126 ± 2 ^b^	129 ± 1 ^b^	129 ± 1 ^b^
	Dry Matter (g)	27 ± 0 ^a^	25 ± 1 ^b^	25 ± 0 ^b^	23 ± 1 ^c^
	Protein (g)	7.0 ± 0.3 ^a^	5.7 ± 0.2 ^b^	4.8 ± 0.1 ^c^	4.0 ± 0.2 ^d^
	Protein Yield (% of Starting Material)	49 ± 0 ^a^	40 ± 0 ^b^	34 ± 0 ^c^	28 ± 0 ^d^
Protein Precipitation After pH Adjustment	**Supernatant**				
	Mass (g)	330 ± 1 ^a^	330 ± 4 ^a^	320 ± 1 ^b^	300 ± 2 ^c^
	Dry Matter (g)	12 ± 0 ^a^	12 ± 1 ^a^	9.5 ± 0.4 ^b^	8.9 ± 0.8 ^b^
	Protein (g)	4.5 ± 0.3 ^a^	4.5 ± 0.1 ^ab^	3.8 ± 0.2 ^b^	2.6 ± 0.2 ^c^
	Protein Yield (% of Starting Material)	32 ± 0 ^a^	31 ± 0 ^ab^	26 ± 0 ^b^	18 ± 0 ^c^
	**Precipitate**				
	Mass (g)	29 ± 1 ^a^	38 ± 1 ^b^	52 ± 2 ^c^	74 ± 3 ^d^
	Dry Matter (g)	6.1 ± 0 ^a^	8.4 ± 0.1 ^b^	11 ± 0 ^c^	13 ± 0 ^d^
	Protein (g)	2.7 ± 0.0 ^a^	4.1 ± 0.1 ^b^	5.7 ± 0.2 ^c^	7.7 ± 0.3 ^d^
	Protein Yield (% of Starting Material)	19 ± 0 ^a^	29 ± 0 ^b^	40 ± 0 ^c^	54 ± 0 ^d^

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
