# Peer review of "Emulsifying and Anti-Oxidative Properties of Proteins Extracted from Industrially Cold-Pressed Rapeseed Press-Cake"

_foods, 2020, doi:10.3390/foods9050678_

Round 1

Reviewer 1 Report

Dear Authors,

the work titled "Emulsifying and anti-oxidative properties of proteins extracted from industrially cold-pressed rapeseed press cake" reports some results on the the emulsifier properties of the rapeseed extract and its apparent antioxidant activity.

Although the topic could be of interest, I am sorry to state that the work seems not studying in details any specific aspect, but just collecting very different results, each trying to answer to broadly different hypothesis. Accordingly, the claim reported does not seem sufficiently supported by the results.

In detail:

1) The extraction technology is not optimized as the samples are just received from a company.

2) The emulsifying properties have been studied only by droplet diameter. Results reported in section 3.2.1 are weakly expressed (no statistics or explicit expression of the results). The effects of pHs and protein concentration are reported only qualitatively. Statistical analysis of the results is missing.

Then, from page 9 line 283 to page 10 line 307, there is a discussion about the results reported by others. But the Authors themselves explain at page 9 lines 282 that any comparison is not possible, since starting materials, extraction methods and formulation are different. So the discussion sounds a little speculative.

Overall, this section (3.2.1) does not provide novel information.

Furthermore, how Authors are sure that the distribution is bimodal? The presence of such big particles (>100 µm) will inevitably obscure the analysis of small particles, thus the fact that the distribution is bimodal could be just a multiple scattering problem.

Without validation of the results, droplet size (d43) in the range of 20-100 µm seems in contrast with the main peak observed in Figure 4b, which fall at 100 µm diameter size.

Morever, fitting quality of the results are not given. Which between Mie theory or Fraunhofer is used for fitting? Which rpm are used during the measurement? Are ultrasounds used? Which dispersant has been used in the cell of the mastersizer? Many parameters of the light scattering measurement are missing or not validated.

In experimental, it is reported that the Author used 0.1% of sodium benzoate to control microbial growth. However, sodium benzoate is also a powerful antioxidant. If this such antioxidant is present in the rapeseed protein sediment, how were you able to measure the antioxidant activity of the proteins in the emulsion?

Finally, the introduction could be improved. In particular, the gaps in knowledge are not well defined. If you can state better what is not known, this would improve the resulting original-sound of your work.

Minor:

TAble 1, raw material could be removed. The results here are just the repetition of the same initial values. You could move them in the Table caption.

Table 1. Check figures of merits.In particular, expression of the results should agree with the digits of the standard deviation.

Author Response

Response to Reviewer #1

Dear Reviewer!

Thanks for the valuable comments. We have done our best to make the suggested changes and hope you find them satisfactory in the revised manuscript. Please find our responses to the comments below. In the revised manuscript new and significantly modified sections are indicated in green.

Dear Authors,

the work titled "Emulsifying and anti-oxidative properties of proteins extracted from industrially cold-pressed rapeseed press cake" reports some results on the the emulsifier properties of the rapeseed extract and its apparent antioxidant activity.

Although the topic could be of interest, I am sorry to state that the work seems not studying in details any specific aspect, but just collecting very different results, each trying to answer to broadly different hypothesis. Accordingly, the claim reported does not seem sufficiently supported by the results.

In detail:

Comment #1: The extraction technology is not optimized as the samples are just received from a company.

Response: We are not sure whether the reviewer is referring to oil extraction technology or protein extraction technology. In a previous study we have investigated the effect of oil extraction technology (cold-pressed, hot-pressed, solvent extracted) and found that the protein yield was significantly higher when cold-pressed rapeseed press cake was used as raw material compared with hot-pressed rapeseed meal or solvent-extracted rapeseed meal (Östbring, K.; Malmqvist, E.; Nilsson, K.; Rosenlind, I.; Rayner, M. The effects of oil extraction methods on recovery yield and emulsifying properties of proteins from rapeseed meal and press cake. Foods 2020, 9, 19.). When it comes to the protein extraction process, we have varied the alkali pH in the leaching phase with cold-pressed rapeseed press cake as starting material in the present manuscript. We have isolated the protein in our lab and the samples were not received from a company. It was only the raw material, the press cake, that was received from a company as was stated on line 127.

Comment #2: The emulsifying properties have been studied only by droplet diameter. Results reported in section 3.2.1 are weakly expressed (no statistics or explicit expression of the results). The effects of pHs and protein concentration are reported only qualitatively. Statistical analysis of the results is missing.

Response: Thanks for the comment, we have performed statistical analysis in SPSS and have rewritten the results section accordingly to the results. We re-analyzed the emulsion stability test with fish oil (Fig 5a) and the oxidation data (Fig 6), in order to use the same statistical method throughout the manuscript (General linear model with Tukeys test, SPSS version 25) (lines 235-240). 

Comment #3: Then, from page 9 line 283 to page 10 line 307, there is a discussion about the results reported by others. But the Authors themselves explain at page 9 lines 282 that any comparison is not possible, since starting materials, extraction methods and formulation are different. So the discussion sounds a little speculative.

Response: We agree that it is difficult, but we still want to refer to other studies and have reported the studies that is most similar to the process used in the present manuscript. We want to communicate to the reader that other researchers also have found emulsifying properties in rapeseed protein. We have added a comment on the value of comparing with other studies to make it more clear for the reader (lines 326-327)

Comment #4: Overall, this section (3.2.1) does not provide novel information.

Response: Section 3.2.1 together with Figure 4 reports how emulsifying properties depends on alkali pH in the leaching phase and that it is probably a matter of changes in tertiary structure that is changing the hydrophilic/hydrophobic regions exposed for the environment. Our opinion is that the study of emulsifying properties is strengthening the manuscript and that the quality would decrease if we removed it.

Comment #5: Furthermore, how Authors are sure that the distribution is bimodal? The presence of such big particles (>100 µm) will inevitably obscure the analysis of small particles, thus the fact that the distribution is bimodal could be just a multiple scattering problem. Without validation of the results, droplet size (d43) in the range of 20-100 µm seems in contrast with the main peak observed in Figure 4b, which fall at 100 µm diameter size.

Response: In a previous study (Östbring, K.; Malmqvist, E.; Nilsson, K.; Rosenlind, I.; Rayner, M. The effects of oil extraction methods on recovery yield and emulsifying properties of proteins from rapeseed meal and press cake. Foods 2020, 9, 19.), we have measured the size of the protein aggregates in the precipitate and found them to be 5-10 µm which fits the range of the small peak in Fig 4c and Fig 4d. We therefore argue that the small peak are protein aggregates not associated to the oil-water interface. Figure 4c in the submitted manuscript shows the volume distribution of emulsions with 0.9-1.1 mg protein/mL oil and droplet size (d43) is 65-92 µm. We understand the reviewers point and in the revised manuscript we now report volume distributions reflecting both 0.9-1.1 mg protein/mL oil (concentration before the plateau in Fig 4a) and 14-18 mg protein/mL oil (far out on the plateau in Fig 4a). We have also added a comment about the volume distributions and how peaks is changing with protein concentration in the emulsions (lines 318-324). We think it is interesting that the small peak around 5 µm has a higher magnitude in the figure below compared to the figure representing 0.9-1.1 mg protein/ml oil which further strengthen the observation that the small peak is protein aggregates not associated to the oil/water interface. When the plateau is reached (in fig 4a in the manuscript) the oil/water interface is saturated with protein and when even more proteins are added, they cannot attach to the interface but remain in the continuous phase.

Figure 1. Volume distribution for emulsions with 0.9-1.1 mg protein/mL oil (left) and 14-18 mg protein/mL oil (right). Left figure is now Figure 4c and right figure is now Figure 4d. 

Comment #6: Morever, fitting quality of the results are not given. Which between Mie theory or Fraunhofer is used for fitting? Which rpm are used during the measurement? Are ultrasounds used? Which dispersant has been used in the cell of the mastersizer? Many parameters of the light scattering measurement are missing or not validated.

Response: We have used the Mie theory with the model “General purpose enhanced sensitivity”. The speed of the stirrer was set to 2000 rpm as was stated in the Material and method section line 186. We have tested different rpm’s and have performed a manual particle size distribution by measuring over 300 droplets for varying stirrer speed under microscope. 2000 rpm fits the manual particle size distribution very well. Lower rpm in our experimental setting cause aggregation and the particle size is too high whereas higher rpm disrupts the droplets reporting too small particle size. Ultrasound was not used, we used an Ystral hand homogenizer to prepare the emulsions. Dispersant was MilliQ water as was stated on line 185. Refractive index for both the sample and continuous phase is reported as well as absorption and obscuration rate (Section 2.4.2).  

Comment #7: In experimental, it is reported that the Author used 0.1% of sodium benzoate to control microbial growth. However, sodium benzoate is also a powerful antioxidant. If this such antioxidant is present in the rapeseed protein sediment, how were you able to measure the antioxidant activity of the proteins in the emulsion?

Response: Thanks for the valuable comment. 0.1% sodium benzoate was only added to stability experiments for emulsions with fish oil (Fig 5a), not the oxidation experiments. The material and method section were very misleading, and we are grateful for the comment. The paragraph describing the addition of sodium benzoate has now been moved to section 2.4.3 “Emulsion stability during storage” where it belongs (lines 193-196). 

Comment #8: Finally, the introduction could be improved. In particular, the gaps in knowledge are not well defined. If you can state better what is not known, this would improve the resulting original-sound of your work.

Response: Thank you for the valuable comment, we have tried to formulate the gaps in knowledge which is that most of the studies on rapeseed protein are performed on hot-pressed rapeseed meal where the proteins are subjected to high temperatures (lines 104-107). We have in a previous study showed that the protein yield from industrially cold-pressed press cake was higher due to lower degree of protein denaturation. To our knowledge, there are no studies on how alkali pH affects yield and emulsifying properties of industrially cold-pressed rapeseed press cake and we have added a paragraph communicating that to the reader (lines 108-110). Also, our opinion is that many studies in the rapeseed protein field fail to report a proper mass balance over the protein through the different processing steps. In fact, yield is calculated in different ways and in many cases the fraction of protein that is solubilized in the leaching step are referred to as a yield which is confusing when other studies are referring to the proteins in the final precipitate as yield. In this manuscript we quantify and report the protein fraction throughout the processes in a structured way to avoid misunderstandings. Furthermore, we have added information about the other proteins in rapeseed and their emulsifying properties to improve the introduction section (lines 57-65 and lines 89-94).

Minor:

Comment #9: Table 1, raw material could be removed. The results here are just the repetition of the same initial values. You could move them in the Table caption.

Response: Thanks for the comment, we agree and have moved the initial values to the table caption (lines 289-290).

Comment #10: Table 1. Check figures of merits. In particular, expression of the results should agree with the digits of the standard deviation.

Response: Thanks for the comment, we have adjusted the significant digits in the manuscript and followed the guidelines in the document below

Reviewer 2 Report

The study presented in this manuscript provides useful information for additng value to rapeseseed press cake. Authors show a way to recover protiens from rapeseed cake and the value of those protiens in oil-awter emulsion stabilization and protecting unsaturated fatty acid-rich oil fration of the emulsion against oxidation.

Following comments will be helpful in revising this manuscript.

Introduction: May require information on storage protiens of rapseed and their contribution to o/w emulsion formation and stabiization.

Materials and Methods:

Miglyol is the trade name of medium chain triglyciride composed of C8 and C10 fatty acids. Better use the chemical description. Where was it obtained from? Was it Migloyl 810? or 812?

 Authors have not described emulsion preparation  with corn or flax oil but present it under results and discussion. Need to provide that information.

Protein extraction: Figure 2 centrifugation steps and Table 1 centrifugation 1 and 2 has to be consistently labelled. Suggestion for Table 1- better use 1st centrifugation as centrifugation after meal extraction and 2nd centrifugation - protein precipitation after pH adjustment. 

The word "precipitate" may be better than "sediment".

Results and Discussion:

Authours have down played the contribution from storage proteins in the precipitate. Looking at the protein yield in the precipitate there may be storage proteins. Wijesundara et al 2013 JFS 78:1340 reference show the extraction pH depence on the composition of recovered protein at pH 6.5. Also Jolivet et al 2011 J Plant Physiol 168: 2015 shows oleosin is about 20 % of total seed protiens at the seed maturity. The  precipitate recovered may have proteins other than oil body proteins. Aren't  there any contaminating oil in the protein product? Discussion needs revison considering these.

Author Response

Response to Reviewer #2

Dear Reviewer!

Thanks for the valuable comments. We have done our best to make the suggested changes and hope you find them satisfactory in the revised manuscript. Please find our responses to the comments below. In the revised manuscript new and significantly modified sections are indicated in green. We feel that the comments have improved the manuscript significantly.

The study presented in this manuscript provides useful information for additng value to rapeseseed press cake. Authors show a way to recover protiens from rapeseed cake and the value of those protiens in oil-awter emulsion stabilization and protecting unsaturated fatty acid-rich oil fration of the emulsion against oxidation. 

Following comments will be helpful in revising this manuscript.

Comment #1: Introduction: May require information on storage protiens of rapseed and their contribution to o/w emulsion formation and stabiization.

Response: Thanks for the comment, we agree and have added a paragraph on rapeseed storage proteins and their emulsifying properties (lines 57-65 and lines 89-93).

Materials and Methods:

Comment #2: Miglyol is the trade name of medium chain triglyciride composed of C8 and C10 fatty acids. Better use the chemical description. Where was it obtained from? Was it Migloyl 810? or 812?

Response: Thanks for the comment, we used Miglyol 812 and the details for purchase is added in the material section (line 123) and changed to “medium chain triglyceride oil” (line 179).

Comment #3: Authors have not described emulsion preparation with corn or flax oil but present it under results and discussion. Need to provide that information.

Response: The information is stated in the paragraph “Oxidative stability of emulsions” in the Material & method section (lines 202-213).

Comment #4: Protein extraction: Figure 2 centrifugation steps and Table 1 centrifugation 1 and 2 has to be consistently labelled. Suggestion for Table 1- better use 1st centrifugation as centrifugation after meal extraction and 2nd centrifugation - protein precipitation after pH adjustment. 

Response: Thanks for the valuable comment, we have changed the labels in Table 1 to be more consistent.

Comment #5: The word "precipitate" may be better than "sediment".

Response: We agree and have changed to “precipitate” throughout the manuscript.

Results and Discussion:

Comment #6: Authours have down played the contribution from storage proteins in the precipitate. Looking at the protein yield in the precipitate there may be storage proteins. Wijesundara et al 2013 JFS 78:1340 reference show the extraction pH depence on the composition of recovered protein at pH 6.5. Also Jolivet et al 2011 J Plant Physiol 168: 2015 shows oleosin is about 20 % of total seed protiens at the seed maturity. The precipitate recovered may have proteins other than oil body proteins. Aren't  there any contaminating oil in the protein product? Discussion needs revison considering these.

Response: We agree, since the protein recovery yield is over 10% for all experiments (Table 1), there must be a co-precipitation of storage proteins as well. We have revised the discussion to reflect this (lines 271-282 and 311-315). Thanks for focusing our attention on this matter. We have not analyzed the oil content in this study but will definitely do that in our future work.

Reviewer 3 Report

I wonder why Authors have chosen MDA-TBA test as an oxidative stability test. It shows many drawbacks. It is non-specific, reaction of MDA with proteins, it is not only end product of fatty acids decomposition to name some of problems with this test. In my opinion the Authors should at least have some comments on those question.

  1. Briefly summarize the content of the manuscript;

This manuscript presents another approach to utilize waste of industrial production of cold-pressed rapeseed press-cake  and apply recovered protein as an emulsifier in food production

Extraction process is well described and might be easily adopted for commercial use.

Scientific part devoted to determine the emulsifying and antioxidative properties is well prepared

and documented.

  1. Illustrate what are, in your opinion, the manuscript’s strengths and weaknesses [this is an essential step, because the Editor will consider the reasoning behind your recommendation and needs to understand it properly];

The use of appropriate methods to carry out experiments

Thorough explanations of observed phenomena indicating thoughtful  understanding of presented problem

  1. Provide a point-by-point list of your major recommendations for the improvement of the manuscript;

Antioxidative properties  determination using MDA-TBARS method is my major objective since this method have many drawbacks regarding exact determination of MDA during oxidation in such complicated system. However, this method has been used consequently for whole investigated systems thus use of this method is acceptable.

Regarding DLS measurements should by also added plot showing number versus diameter, since presence of big structures even in small number may overwhelm small sizes structures

  1. If necessary, provide a point-by-point list of your minor for the improvement of the manuscript.

maybe in line 142 in definition should be added word:  final sediment

In general the manuscript is interesting, very good experimental work, well written, and ready to be publish.

Author Response

Response to Reviewer #3

Dear Reviewer!

Thanks for the valuable comments. We have done our best to make the suggested changes and hope you find them satisfactory in the revised manuscript. Please find our responses to the comments below. In the revised manuscript new and significantly modified sections are indicated in green. We feel that the comments have improved the manuscript significantly.

Comment #1: I wonder why Authors have chosen MDA-TBA test as an oxidative stability test. It shows many drawbacks. It is non-specific, reaction of MDA with proteins, it is not only end product of fatty acids decomposition to name some of problems with this test. In my opinion the Authors should at least have some comments on those question.

Response: Thanks for the comment. We are aware of the limitations with TBARS: it is a non-specific method, MDA’s can react with other MDA’s as well as with proteins which can give either false positive reponse or false negative response. MDA’s are not the sole end product of fatty peroxide formation and MDA can be generated also from other reactions. Further, since TBARS is a method based on photospectrometry, different substances with similar structure can give rise to a signal independent of origin. On the other hand, it is a method widely used although it gives a limited view of the complex lipid peroxidation process. In the interpretation of the results we have carefully avoided to draw any conclusions on concentrations, only compared the overall oxidative stability of the different emulsions in broad terms. We measured MDA in all four types of emulsions every second day to make sure that we could follow the oxidation. First, we did experiments with a flaxseed oil bought in the supermarket, but it turned out that the oil was already fully oxidized. We therefore went to a farm one hour from our university and pressed flaxseed ourselves, we then drove back to the university and prepared all emulsions the same day as the oil was pressed. This time we managed to follow the oxidation in the oil. We have commented the method and added details around the flaxseed oil (lines 215-216 and 204-206). 

[Briefly summarize the content of the manuscript]:

This manuscript presents another approach to utilize waste of industrial production of cold-pressed rapeseed press-cake  and apply recovered protein as an emulsifier in food production

Extraction process is well described and might be easily adopted for commercial use.

Scientific part devoted to determine the emulsifying and antioxidative properties is well prepared 

and documented.

[Illustrate what are, in your opinion, the manuscript’s strengths and weaknesses [this is an essential step, because the Editor will consider the reasoning behind your recommendation and needs to understand it properly];

The use of appropriate methods to carry out experiments

Thorough explanations of observed phenomena indicating thoughtful  understanding of presented problem 

[Provide a point-by-point list of your major recommendations for the improvement of the manuscript;]

Comment #2: Regarding DLS measurements should by also added plot showing number versus diameter, since presence of big structures even in small number may overwhelm small sizes structures

Response: We used light scattering (Malvern Mastersizer 2000), not dynamic light scattering. The particle size distributions shown in Fig 4 shows a frequency distribution of d43 where  d43=. In emulsion research the d43 is the most common particle size variable because it is the most indicative of the creaming phenomenon which is one of the most common destabilization mechanisms in emulsions. We used the d32, the surface weighted mean, to scale the surfaces of emulsions droplets in the antioxidant experiments. In the context of this formulation, providing a number mean would misrepresent the emulsion due to the fact that we have small protein aggregates which would dominate the number of particles in the emulsion. However, it has very little physical impact on the emulsions’ overall microstructure.    

[If necessary, provide a point-by-point list of your minor for the improvement of the manuscript.]

Comment #3: maybe in line 142 in definition should be added word:  final sediment

Response: Thanks for the comment, according to another reviewer we have changed “sediment” to “precipitate” and have now included “final” to further increase the clarity of the equation (line 156).

In general the manuscript is interesting, very good experimental work, well written, and ready to be publish.

Round 2

Reviewer 2 Report

Revised manuscript contains additional information and has addressed the conerns raised by the reviwers.